# Content, Structure and Delivery Characteristics of Yoga Interventions for the Management of Osteoarthritis: A Systematic Review Protocol

**DOI:** 10.3390/ijerph19105806

**Published:** 2022-05-10

**Authors:** Isha Biswas, Sarah Lewis, Kaushik Chattopadhyay

**Affiliations:** 1Lifespan and Population Health Academic Unit, School of Medicine, University of Nottingham, Nottingham NG5 1PB, UK; sarah.lewis@nottingham.ac.uk (S.L.); kaushik.chattopadhyay@nottingham.ac.uk (K.C.); 2The Nottingham Centre for Evidence-Based Healthcare: A JBI Centre of Excellence, Nottingham NG5 1PB, UK

**Keywords:** osteoarthritis, management, yoga, systematic review

## Abstract

The global burden of osteoarthritis among adults is rising. Yoga might be a potential solution for the management of osteoarthritis. This systematic review aims to synthesise the content, structure and delivery characteristics of effective yoga interventions for the management of osteoarthritis. The JBI methodology for systematic reviews of effectiveness and the Preferred Reporting Items for Systematic Reviews and Meta-analyses (PRISMA) guidelines will be followed. Randomised controlled trials (RCTs) assessing the effectiveness of yoga interventions for the management of osteoarthritis in adults will be included in this review. We aim to search the following databases to find published and unpublished studies: MEDLINE, EMBASE, CINAHL, PsycInfo, SPORTDiscus, AMED, Web of Science, CENTRAL, TRIP, AYUSH Research Portal, ABIM, CAM-QUEST, PeDro, OpenGrey, EthOS, ProQuest Dissertations and Theses and DART-Europe-e-theses portal. No date or language restrictions will be applied. A narrative synthesis will be conducted with the help of tables. A meta-regression will be conducted to explore the statistical evidence for which the components (content, structure and delivery characteristics) of yoga interventions are effective.

## 1. Introduction

### 1.1. Osteoarthritis

Osteoarthritis is a long-term condition of the joints with symptoms such as pain, stiffness and difficulty in movement [1]. It is the most common type of arthritis in the world [1,2]. It erodes the cartilage lining of the joints, causing joint deformities [1]. Traditionally, it is considered to be “non-inflammatory” [2]. However, evolving research has found inflammatory mediators to be involved in the onset and progress of the condition, making it difficult to classify osteoarthritis as inflammatory or non-inflammatory [3]. It can affect any joint, but the most common ones affected are the knees, hips and small joints of the hands [1]. The diagnosis of osteoarthritis is based on physical examination, radiographic and magnetic resonance imaging (MRI) findings and/or arthroscopy [1,4]. Radiographic findings do not always correlate with the clinical severity of osteoarthritis [5]. Some of the comorbidities associated with osteoarthritis are diabetes, cardiovascular diseases, hypertension, metabolic syndrome, stroke, back pain and depression [6,7,8].

### 1.2. Types and Risk Factors of Osteoarthritis

Osteoarthritis can be classified into two types—primary and secondary [2]. Primary osteoarthritis is the most common type and is not caused by a pre-existing health condition but is associated with some risk factors [2]. The risk factors for osteoarthritis include sociodemographic factors, such as increasing age, female sex and ethnicity (globally, Black/African-American people are more prone to knee osteoarthritis compared to Non-Hispanic White/Caucasian people); lifestyle factors such as a lack of physical activity; genetic factors; health conditions, such as obesity, muscle weakness and joint injury (from physically strenuous occupation and sports activities); and nutritional deficiencies, such as a deficiency of vitamin D, vitamin C and vitamin K [1,9,10,11]. Secondary osteoarthritis occurs due to a pre-existing health condition, including but not limited to, joint abnormality, any other type of arthritis (such as rheumatoid arthritis and gout) and osteoporosis [1,2].

### 1.3. Global Burden of Osteoarthritis

Osteoarthritis poses tremendous health (physical and psychological), social and economic burden [12,13,14,15,16,17,18]. Some of the physical health consequences of osteoarthritis include its symptoms, such as joint pain and stiffness, difficulty in movement and occurrences of falls [13,14]. Some common psychological health consequences are anxiety and depression [12]. Social consequences include withdrawal from social participation and occupational activities [13]. The economic impact of osteoarthritis includes huge direct and indirect costs [14,16]. Direct costs arise due to medical and surgical treatments such as knee replacement surgery [17,18]. Indirect costs arise due to absenteeism from work and the loss of work productivity [16]. Reduced self-efficacy and poor quality of life are some other issues [15].

In terms of numbers and percentages, about 3.3–3.6% of the global population is affected by osteoarthritis, and it causes moderate-to-severe disability, making it the 11th most debilitating disease in the world [2]. Osteoarthritis contributes to 18.9 million years lived with disability (YLDs), which is 2.2% of the total global YLDs [2]. Osteoarthritis of the knee, hip, hand and other joints (e.g., foot, shoulder and wrist) contribute to 60.9%, 5.5%, 23.5% and 10.2% of global YLDs, respectively [2]. The global disability-adjusted life years (DALYs) related to osteoarthritis are high and rising [19].

### 1.4. Current Management Strategies of Osteoarthritis and Their Limitations

The main aim of osteoarthritis management is to minimise joint pain and the loss of function [20]. It is managed using pharmacological and non-pharmacological interventions, as well as surgical interventions in severe cases [20]. Pharmacological interventions include oral, topical and/or intraarticular options [2]. Non-steroidal anti-inflammatory drugs (NSAIDs), prescribed orally (e.g., ibuprofen), are the first line of treatment but have side effects, such as gastrointestinal toxicity and cardiovascular effects [2]. Topical NSAIDs (e.g., diclofenac gel) are less effective than their oral counterparts and have fewer gastrointestinal and other systemic side effects; however, they often cause local skin irritation [21]. Intra-articular corticosteroid injections provide short-term relief from pain and improve function, but using them more than once every four months can result in cartilage and joint damage [22]. The side effects and costs associated with the use of pharmacological interventions are some of the reasons why people with osteoarthritis might be reluctant to use them [20,22]. Along with pharmacological interventions, the use of non-pharmacological interventions, such as exercise to improve muscle strength (e.g., the hamstring muscle in the case of knee osteoarthritis) and weight loss for overweight and obese individuals, are recommended [23]. Surgical treatments such as joint replacements are needed when other treatments have not been effective or in cases of severe joint damage [2].

### 1.5. Yoga: A Potential Solution for the Management of Osteoarthritis

Yoga is an ancient practice, with origins in the Indian subcontinent, that aims to offer a holistic sense of well-being of the body and mind [24]. Yoga philosophy and practice were first described by Patanjali in the classic text *Yoga Sutras* [25]. The multi-factorial approach of yoga includes components such as yogic poses (asana), breathing practices (pranayama) and meditation (dhyana) and relaxation practices, along with moderation in lifestyle [25]. Among the seven major branches of yoga, Hatha yoga is the most popular [26]. There are various styles of Hatha yoga, and each has its distinct emphasis on the individual components [26]. Yoga is becoming popular across the world, and there are about 300 million people who practice yoga globally [27]. Generally, yoga is easy to learn and safe to practice, demands a low-to-moderate level of supervision, is inexpensive to maintain because of the minimal equipment requirement and can be practised indoors and outdoors [28,29]. 

Several systematic reviews and meta-analyses have reported the beneficial effects of yoga interventions on osteoarthritis outcomes, such as pain relief and functional improvement [30,31,32,33,34,35]. These reviews have included randomised controlled trials (RCTs) [30,31,32,33,34], except one, which also included other study designs [35]. This recent systematic review and meta-analysis of 20 RCTs and 2 case series on knee and hip osteoarthritis showed that yoga significantly improved pain (mean difference (MD) −1.82, 95% confidence interval (CI) −2.96 to −0.67) and physical function (−6.07, −9.75 to −2.39) compared to no intervention or usual care [35]. No adverse events related to yoga were reported [32]. 

The beneficial effects of yoga on osteoarthritis outcomes can be explained by some potential mechanisms. Yoga practice generally begins with slow movement sequences to increase blood flow and warm up the muscles [36]. This is followed by holding certain yogic poses, including flexion, extension, adduction, abduction and rotation, which engage the muscles in isometric contraction [36,37,38]. The movement of joints increases flexibility, whereas standing yogic poses improve balance and coordination by strengthening major muscle groups (e.g., hamstring muscles and quads) [39,40,41]. This might lead to a reduction in pain and stiffness and improved function [41,42,43]. There is some evidence that shows that the practice of specific yogic poses and breath control practices improve the process of respiration and calm the mind by reducing stress, anxiety and depression [43,44,45]. That is, yoga potentially provides physical health benefits; reduces stress, anxiety and depression; and enhances self-esteem and quality of life [30]. 

### 1.6. The Rationale for the Systematic Review

All the above-mentioned systematic reviews have only described but not synthesised the content, structure and delivery characteristics of yoga interventions for the management of osteoarthritis [30,31,32,33,34,35]. Moreover, a meta-regression has never been conducted to explore the statistical evidence for which the components of yoga interventions (e.g., content, structure and delivery characteristics) are effective. Thus, there is a need to conduct such a systematic review so that the content, structure and delivery characteristics of effective yoga interventions for the management of osteoarthritis can be synthesised and used in future research and practice.

### 1.7. Aim

The aim of this systematic review is to synthesise the content, structure and delivery characteristics of effective yoga interventions for the management of osteoarthritis.

## 2. Methods

This systematic review will be conducted in accordance with the JBI methodology for systematic reviews of effectiveness and the Preferred Reporting Items for Systematic Reviews and Meta-analyses (PRISMA) guidelines [46,47]. The systematic review protocol is registered with PROSPERO (CRD42022298155).

### 2.1. Inclusion Criteria

#### 2.1.1. Population

This systematic review will include studies conducted among adults (aged ≥18 years) diagnosed with osteoarthritis of one or more joints. No restrictions will be applied regarding the diagnostic criteria of osteoarthritis, and to name a few, diagnoses based on physical examination, radiographic and MRI findings and/or arthroscopy will be included. If a study includes both children and adults, only the relevant information about adults will be extracted. If it is not possible to extract the relevant information about adults, the study will be excluded.

#### 2.1.2. Intervention

Studies reporting at least one of the major components of yoga, namely, asana (yogic poses), pranayama (breathing practices) and dhyana (meditation) and relaxation practices, will be included. There will be no restrictions on the type, frequency, duration and delivery mode of the yoga intervention. Studies that include multimodal interventions (which include yoga among other interventions) will be excluded if relevant data cannot be extracted. Studies will also be excluded if they did not explicitly label the intervention as yoga.

#### 2.1.3. Comparator

Studies comparing yoga interventions with no intervention, sham intervention, non-pharmaceutical intervention (e.g., diet, physical activity and educational intervention) or pharmaceutical intervention (e.g., NSAIDs) will be included. Studies with co-interventions will be included as long as all the eligible study groups were allowed to do so. Studies with a head-to-head comparison of two or more yoga interventions (i.e., different in terms of content, structure or delivery characteristics) will be excluded.

#### 2.1.4. Outcome

This systematic review will include studies that assessed the core outcomes of osteoarthritis, i.e., pain and function, as recommended by several guidelines [48,49,50,51,52]. Pain assessed using any scale will be eligible (e.g., visual analogue scale (VAS) and numeric rating scale (NRS)), and function assessed using any scale will be eligible (e.g., arthritis impact measurement Scale (AIMS), any joint-specific scale such as Foot and Ankle Ability Measure (FAAM), Knee Injury and Osteoarthritis Outcome Score (KOOS) and Hip Disability and Osteoarthritis Outcomes Survey (HOOS)) [51,52]. Radiographic outcomes, such as joint space narrowing and osteocyte formation, will not be included in this review, as the inter- and intra-observer variabilities in interpreting radiographs may affect the specificity of the classification criteria and are not standardised to be considered as core outcomes [48,51].

#### 2.1.5. Study Design

Considering the feasibility and practicality of the proposed work and the hierarchy of study designs, only RCTs will be included in this systematic review.

### 2.2. Data Sources and Search Strategies

The following 13 databases will be searched from their inception dates to find published studies: (i) MEDLINE (from 1946; Ovid), (ii) EMBASE (from 1974; Ovid), (iii) CINAHL (from 1994; EBSCOHost), (iv) PsycInfo (from 1806; Ovid), (v) SPORTDiscus (from 2004; EBSCOhost), (vi) Allied and Complementary Medicine (AMED) (from 1985; Ovid), (vii) Web of Science (from 1900; Clarivate analytics), (viii) Cochrane Central Register of Controlled Trials (CENTRAL) (from 1996), (ix) Turning Research Into Practice (TRIP) (from 2014), (x) AYUSH Research Portal (http://ayushportal.nic.in/, accessed on 21 January 2022), (xi) A Bibliography of Indian Medicine (ABIM) (http://indianmedicine.eldoc.ub.rug.nl/, accessed on 21 January 2022), (xii) CAM-QUEST (https://www.cam-quest.org/en, accessed on 21 January 2022) and (xiii) Physiotherapy Evidence Database (PeDro) (from 1999). Unpublished studies will be searched using (i) OpenGrey (from 1997), (ii) EthOS (from 1925), (iii) ProQuest Dissertations and Theses (from 1980) and (iv) DART-Europe-e-theses portal (from 1999). The reference list of all the included studies and relevant previous systematic reviews will be screened for additional studies.

The search strategies are developed based on the following and in consultation with a Research Librarian at the University of Nottingham: (i) the yoga component is based on a previous relevant systematic review protocol [53], (ii) the osteoarthritis component is based on the search strategies reported in the UK’s National Institute for Health and Care Excellence (NICE) guidelines for the management of osteoarthritis [54] and existing Cochrane systematic reviews on osteoarthritis [55,56], and (iii) the pre-designed search filters for RCTs are used [57,58,59]. All the search strategies are detailed in Appendix A. No date or language restrictions will be applied.

### 2.3. Study Screening and Selection

All the identified citations will be collated and uploaded onto Endnote X9 (Clarivate Analytics, PA, USA) [60], and duplicates will be removed. The remaining records will then be imported into Rayyan (Qatar Computing Research Institute (Data Analytics), Doha, DH, Qatar) [61] to facilitate the title and abstract screening process. Titles and abstracts will be independently screened for eligibility using the inclusion criteria by two systematic reviewers (I.B. and S.L./K.C.). Studies identified as potentially eligible or those without an abstract will have their full text retrieved. The full texts of the studies will be assessed for eligibility by two independent reviewers. Full-text studies that do not meet the inclusion criteria will be excluded, citing reasons. Any disagreements that arise between the two reviewers will be resolved through discussion. If consensus is not reached, a third reviewer will be consulted. Translations will be sought where necessary.

### 2.4. Assessment of Methodological Quality

The included studies will be critically assessed by two independent systematic reviewers (I.B. and S.L./K.C.) for methodological quality using the standardised critical appraisal tool developed by JBI for RCTs [46]. This tool uses a series of criteria that can be scored as being met (yes), not met (no), unclear or not applicable (n/a). The two reviewers will independently assess each criterion and comment on it. Any disagreements that arise between the two reviewers will be resolved through discussion. If consensus is not reached, then a third reviewer will be involved. All studies, regardless of their methodological quality, will undergo data extraction and synthesis, where possible.

### 2.5. Data Extraction

Two systematic reviewers (I.B. and S.L./K.C.) will independently extract data from the included studies using a pre-developed and pre-tested data extraction form. Any disagreements that arise between the two reviewers will be resolved through discussion. If consensus is not reached, a third reviewer will be consulted. The following data will be extracted: author(s), year of publication, country, participant characteristics (e.g., age, sex, ethnicity, occupation and type of joints affected by osteoarthritis), sample size, intervention and comparator, outcomes (i.e., pain and function measurements), the timing of follow-up at the end of the intervention and adverse events. For both the outcomes, the authors will extract the end of intervention data [51,62]. Where this time point is not reported, data from the time point closest to the end of the intervention will be extracted. Intention-to-treat (ITT) data will be preferred compared to per-protocol data. ITT analysis is the most preferred analysis method for RCTs [63]. This is because it preserves sample size by including all the participants irrespective of adherence to the study and attrition, maximises external validity and helps in understanding real-world circumstances encountered in an actual setting [63,64]. Post-intervention data will be extracted in preference to change from baseline data (i.e., post-intervention score–baseline score). Percentage change from baseline will not be extracted, as it is highly sensitive to change in variance, and it also fails to protect from baseline imbalances, leading to non-normally distributed outcome data [65]. In addition, the content of yoga interventions will be extracted (e.g., yogic poses, breathing practices, meditation and relaxation practices), along with the structure (e.g., duration of the yoga sessions, and duration and frequency of the yoga interventions) and delivery characteristics (e.g., individual or group sessions, supervised or unsupervised sessions, sessions delivered in yoga centres or at home, strategies for yoga intervention uptake and adherence, and characteristics of yoga instructors).

To obtain missing data on outcomes, multiple strategies will be used. The first strategy will be to contact the corresponding author of the included study by email (at least two times per author) to obtain the relevant data. If we receive no response from the corresponding author, then certain assumptions will be applied. For example, where pain and function are reported as continuous outcomes, if the standard deviation (SD) is missing, SD will be imputed from a similar study (in terms of intervention, comparator, sample size and numerical outcome data) [66]. If only a median and interquartile range (IQR) is reported, these will be extracted, the mean will be assumed to be equal to the median, and the SD will be calculated using the standard formula (=IQR/1.35) [66]. 

### 2.6. Data Synthesis

A narrative synthesis will be conducted with the aid of tables and text, focusing on the content, structure and delivery characteristics of effective yoga interventions for the management of osteoarthritis (i.e., for each type of joint and outcome). For example, in the case of knee osteoarthritis, a narrative synthesis will be performed for knee pain and knee function.

Considering the errors in how authors analyse and report yoga interventions to be effective in studies (e.g., conducting pre-post analysis of outcomes within study arms but no comparative analysis between study arms), meta-analyses will be conducted for each type of joint and outcome using Review Manager 5.4.1 (Copenhagen, The Nordic Cochrane Centre, The Cochrane Collaboration) [67] to determine the true effectiveness of each included yoga intervention. Random-effects meta-analyses will be conducted to provide a weighted measure of treatment effect. For studies with more than one comparator group, the comparisons will be included in separate meta-analysis models to avoid the issue of double-counting of the comparator group. Where pain and function are reported as continuous outcomes, MDs with 95% CIs will be reported where the same scale is used across the studies. Where different scales are used across studies, standardised mean differences (SMDs) with 95% CIs will be reported. Where necessary, post-intervention data will be pooled with changes from baseline data, and this will be carried out for MDs but not SMDs. If reported as binary outcomes, risk ratios with 95% CIs will be reported.

In the final step, a meta-regression will be conducted to explore the statistical evidence for which the components of the intervention are effective [68]. This requires a reasonable number of studies in order to have sufficient power to show differences in effectiveness between components, and so the final decision on which components will be explored and how they will be grouped will be made once we have extracted data from the included studies on the components of each intervention. However, we anticipate exploring broad categories of intervention components, including content (e.g., yogic poses, breathing exercises, meditation and relaxation practices), structure (e.g., number of yoga sessions per week and length of the yoga sessions) and delivery characteristics (e.g., one-to-one or group sessions) of yoga interventions [68]. A random-effects model will be used to analyse these subgroup effects [69]. The effects of content, structure and delivery components on outcomes, i.e., pain and function, will be investigated by looking at the amount of heterogeneity explained by these components, using the reduction in the I^2^ statistic and the DerSimonian–Laird estimation method [70]. The results will establish the statistical significance of any observed patterns in which the components of yoga are associated with a greater effect on osteoarthritis outcomes.

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
