# Peer review of "Content, Structure and Delivery Characteristics of Yoga Interventions for the Management of Osteoarthritis: A Systematic Review Protocol"

_ijerph, 2022, doi:10.3390/ijerph19105806_

Round 1

Reviewer 2 Report

  1. A systematic review and meta-analysis on yoga and osteoarthritis have been published. Can authors justify the difference between their approach and this paper? doi: 10.1007/s11926-019-0846-5.
  2. "the aim of this systematic review is to synthesise the content, structure and delivery characteristics of effective yoga interventions for managing osteoarthritis" It sounds like a meta-synthesis rather than meta-analysis to me. Can the authors reframe the aim so that it is more like a meta-analysis?
  3. 2.1.4: What about radiographic outcomes of OA? Will they be considered?
  4. Can the authors explain why "Intention-to-treat (ITT) data will be preferred compared to per-protocol data"?
  5. When numerical information is missing, the authors should contact the authors of the original studies first before applying the assumptions, which might not be valid. For example, it is obvious that when skewness occurs and the median is used to present the data, it will not be the same as the mean. Median=mean occurs only in normally distributed data.
